# OxKBC: Outcome Explanation for Factorization Based Knowledge Base Completion

**Yatin Nandwani**                              YATIN.NANDWANI@CSE.IITD.AC.IN

**Ankesh Gupta** *                        ANKESHGUPTA.CSEIITD@GMAIL.COM

**Aman Agrawal** *                                AMAN71197@GMAIL.COM

**Mayank Singh Chauhan**        MAYANK.SINGH.CHAUHAN.CS516@CSE.IITD.AC.IN

**Parag Singla**                               PARAGS@CSE.IITD.AC.IN

**Mausam**                                  MAUSAM@CSE.IITD.AC.IN

*Department of Computer Science and Engineering*
*Indian Institute of Technology Delhi, INDIA*

## Abstract

State-of-the-art models for Knowledge Base Completion (KBC) for large KBs (such as FB15k and YAGO) are based on tensor factorization ($TF$), e.g, DistMult, ComplEx. While they produce good results, they cannot expose any rationale behind their predictions, potentially reducing the trust of a user in the outcome of the model. Previous works have explored creating an inherently explainable model, e.g. Neural Theorem Proving (NTP), DeepPath, MINERVA, but explainability in them comes at the cost of performance. Others have tried to create an auxiliary explainable model having high fidelity with the underlying TF model, but unfortunately, they do not scale well to large KBs. In this work, we propose OxKBC – an **O**utcome e**X**planation engine for **KBC**, which provides a post-hoc explanation for every triple inferred by a (uninterpretable) factorization based model. It first augments the underlying Knowledge Graph by introducing weighted edges between entities based on their similarity given by the underlying model. It then defines a notion of human-understandable explanation paths along with a language to generate them. Depending on the edges, the paths are aggregated into second–order templates for further selection. The best template with its grounding is then selected by a neural selection module that is trained with minimal supervision by a novel loss function. Experiments over Mechanical Turk demonstrate that users overwhelmingly find our explanations more trustworthy compared to rule mining.

## 1. Introduction

Knowledge bases (KBs) store facts in the form of triples $r(s, o)$, which indicate that the head entity $s$ is related to the tail entity $o$ via the relation $r$, e.g., $capitalOf(WashingtonDC, USA)$. These KBs are typically incomplete [Bollacker et al., 2008] and the task of KB Completion (KBC) aims to infer new facts from a given KB.

Many state-of-the-art models for KBC on large-scale KBs use tensor factorization ($TF$) [Sedghi and Sabharwal, 2018, Lacroix et al., 2018, Jain et al., 2018, Xue et al., 2018]. They learn embeddings for each entity and relation in a continuous vector space and define a scoring function to evaluate the validity of a given triple. At test time, given a query $r(s, ?)$ they output a ranked list of entities $o$ by sorting the model scores for $r(s, o)$ for all possible tail entities. Though $TF$ models achieve

---

*. Joint second author.

excellent task performance, they are inherently opaque – it is very difficult to uncover a rationale for why the model scores an entity high for a query, because neither the dimensions nor the scoring function are human interpretable. Since users are often the final consumers of a KB (e.g., via a KB-based QA task, or an entity retrieval task [Xiao et al., 2016]), we believe providing a rationale behind a prediction to them is an important sub-task for building trust in these systems.

**Contributions:** We present OXKBC: a post-hoc **O**utcome **eX**planation engine, which is faithful to a given $TF$-based **KBC** model ($M$). It works on the underlying Knowledge Graph, $KG$, corresponding to the facts given in the Knowledge Base, $KB$. It first augments the $KG$ with weighted edges between entities based on their similarity as defined by the underlying model $M$. It then posits that an explanation for a prediction $o$ for query $r(s, ?)$ is a path between $s$ and $o$ in this augmented graph.

A key contribution of our work is that similar paths are grouped into second-order templates and OXKBC trains a neural template selection module for explaining a given prediction. We define novel unsupervised and semi-supervised loss functions, so that this module can be trained with no or minimal training data. OXKBC outputs the highest scoring path within the selected template as an explanation to the user. We evaluate OXKBC explanations via a user study over Amazon Mechanical Turk. We find that workers overwhelmingly prefer OXKBC explanations compared to rule-mining techniques proposed earlier.

## 2. Related Work

**KBC:** Other than $TF$ based models, research has also explored path based approaches for KBC. They use random walk with restarts [Lao and Cohen, 2010] and more recently, reinforcement learning [Das et al., 2018, Xiong et al., 2017] to construct multi-hop paths in the knowledge graph and use those to make predictions. The paths may offer natural interpretability [Nickel et al., 2016], but the performance of these models is significantly lower than $TF$, especially on large KBs like FB15K [Bordes et al., 2013], and YAGO [Suchanek et al., 2007]. See Appendix A.1 for a detailed comparison. [Rocktäschel and Riedel, 2017] inductively learn first order rules to generate paths by using a differentiable version of backward chaining. Unfortunately, it does not scale beyond 10K facts in KB.[1] In contemporaneous work, [Minervini et al., 2020] speed up neural theorem proving for this task. Unfortunately, we could not get their method to complete even after a week of continuous computation on FB15K, suggesting that their method still is not scalable to large KBs. Earlier, research had also considered symbolic Horn-clause style inference [Lao and Cohen, 2010, Schoenmackers et al., 2010], though these are no longer competitive with modern neural methods.

$TF$ methods, like DistMult [Yang et al., 2015] and ComplEx [Trouillon et al., 2016], calculate the score $S^M(s, r, o)$, of a triple $r(s, o)$ via a latent tensor factorization over entity embeddings $(\overline{s}, \overline{t})$ and relation embeddings $(\overline{r})$. Key idea in these methods is to learn embeddings such that the score $S^M$ of the facts present in the $KB$ is high. In this case, DistMult [Yang et al., 2015] aligns relation embeddings to the Hadamard product of the embeddings of head and tail entitites. It uses $S^M(s, r, o) = \overline{r}^T(\overline{s} \bullet \overline{o})$, where $\bullet$ is the element-wise product, also known as Hadamard product.

While in principle OXKBC may be adapted to explain any $TF$ model's predictions, in our experiments, we use TypedDistMult [2] [Jain et al., 2018] as our underlying model $M$. The key idea is to augment the scoring function used by DistMult to reduce type errors. Two new terms are introduced to account for head and tail entity type compatibility. The approach requires no explicit

---

1. https://www.akbc.ws/2017/slides/sebastian-riedel-slides.pdf
2. Code taken from https://github.com/dair-iitd/KBI

specification of type hierarchies or type supervision. The scoring function $S^M$ scores $r(s,o)$ as $S^M(s,r,o) = \sigma(\overline{r}_w^T(\overline{s}_w \bullet \overline{o}_w))\sigma(\overline{r}_{ts}^T\overline{s}_t)\sigma(\overline{r}_{to}^T\overline{o}_t)$, where $\sigma$ is the sigmoid function. The subscript $\underline{w}$ denotes the world knowledge embedding, $\underline{t}$ denotes the type embedding of an entity and $\underline{ts}$ and $\underline{to}$ denote head and tail type embeddings of a relation, respectively.

**Explainability in KBC:** There are broadly two approaches for explainable AI. One is to design a transparent model, which can inherently reveal its functioning (e.g., linear models, decision trees, Horn-clause inference for KBC). Unfortunately, transparent models often result in compromised performance in general [Weld and Bansal, 2018], as well as specifically for KBC (as discussed earlier in this section). An alternative is to provide post-hoc explanations for the behaviour of a high-quality black box model [Wachter et al., 2017]. This approach is further divided into two categories: 'Model Explanation' (Mx) and 'Outcome Explanation' (Ox) [Costabello et al., 2019].

In Mx, once the black–box model is learnt, a global explainable auxiliary model is built using the predictions of the black-box model such that the auxiliary model has high fidelity with the original black-box model. Rule mining [Yang et al., 2015] is an Mx approach, which mines first order rules based on embeddings learnt by $M$. It identifies paths (of length $\leq 3$) using the notion of support and confidence borrowed from the frequent itemset mining literature [Pei et al., 2000]. Search space is pruned by considering only those relations which are close to each other in the embedding space. The extracted first order rules may be used to explain a prediction. We compare against this approach in our experiments. The other Mx approach is known as the pedagogical approach – it trains a global auxiliary interpretable model (weighted Horn clause rules) from the output of the non-interpretable but accurate $TF$ model [Gusmão et al., 2018]. Its experiments were on a toy dataset with only 13 relations, and unfortunately, we were unable to scale this to the size of our dataset (see Appendix A.2).

Ox, attempts to achieve a higher fidelity between explainable model and $M$ by creating local model approximations around each prediction independently, e.g. [Ribeiro et al., 2016]. This approach often trains a new classifier for each prediction, which is slow if a large number of predictions need to be explained. In response, OxKBC takes a novel Ox approach, which decomposes the explanation engine into two components: a global trained model that, for a given prediction, selects the most plausible second-order explanation template (which quantifies over both relations and entities), and an explanation generator that outputs the best instantiation of the template for this prediction. This approach scales better, since the trained model is global and easy to train, and no retraining is required per prediction.

Finally, a related task to Ox is fact checking – verifying the truthfulness of a (predicted) triple $r(s,o)$ based on existing facts in the KB. Many fact-checking approaches [Shiralkar et al., 2017, Shi and Weninger, 2016] use path based techniques to check the veracity of a triple and generate paths as evidence. However, these methods do not use algorithms that use information from the underlying TF model. Thus, while these paths may be useful evidence, they should not be used as explanations, as they are not faithful to the TF model. In the future, it may be interesting to combine fact checking ideas with statistical information in TF models to develop a better outcome explanation engine.

## 3. OxKBC: The Outcome Explanation Engine

We are given a knowledge base of facts, $KB$, with relations $R$ and entities $E$, and a model $M$, trained on $KB$. $M$ scores each triple $r(s,o)$ using the score $S^M(s,r,o)$. The goal of OxKBC is to

show an end-user a reason why $M$ predicts a tail entity $o$ for the query $r(s, ?)$. It does so by finding an explanation path between $s$ and $o$ which entails the given relation $r$.

### 3.1 Language for generating Explanation Paths

A KB can be seen as a knowledge-graph $KG$, where there is a directed edge from $s$ to $o$ labeled by $r$ for every KB fact $r(s, o)$. We now define the notion of an explanation path in this $KG$. Once the model $M$ is learnt, we augment $KG$ with additional edges that correspond to similarity between any two entities $e$ and $e'$. We represent such an edge as $\sim_M (e, e')$ and call them 'entity similarity edges'. Here, the operator $\sim_M$ represents that $M$ considers these two entities similar. We call the original edges of $KG$ as 'relation edges'. An explanation path, $P(s, o)$, is a path between two entities, $s$ and $o$, in the augmented $KG$. Here, we call $s$ as $Head(P)$ and $o$ as $Tail(P)$. A path is a sequence of edges concatenated by a ",". Following grammar generates an explanation path recursively:

1. **Relation Edges:** $P \leftarrow r(s, o) \quad \forall \, r(s, o) \in KB$
2. **Pre-fix a similarity edge:** $P \leftarrow \sim_M (s', s), P_1 \ s.t. \ Head(P_1) = s$.
3. **Post-fix a similarity edge:** $P \leftarrow P_1, \sim_M (o, o') \ s.t. \ Tail(P_1) = o$.
4. **Concatenation of two paths:** $P \leftarrow P_1, P_2 \quad s.t. \ Tail(P_1) = Head(P_2)$.

$P_1$ and $P_2$ on the R.H.S. themselves denote explanation paths generated using the same rules recursively. An explanation path may contain both types of edges: 'relation edges' and 'entity similarity edges'. For each entity similarity edge between two entities $e$ and $e'$, we may define its weight as $sim_M(e, e')$ where $sim_M$ function captures the degree of similarity between the entities as given by the underlying model $M$. This is easy to obtain from any $TF$ model, e.g., for TypedDistMult [Jain et al., 2018], we use $cos(\overline{r}_w, \overline{r}'_w)cos(\overline{r}_{ts}, \overline{r}'_{ts})cos(\overline{r}_{to}, \overline{r}'_{to})$, where $cos$ is the cosine similarity. For a relation edge, we consider its weight as 1 as we include relation edges for only those triples that exist in KB. Another alternative is to use the model score $S^M(s, r, o)$ here as well, like similarity edges, and that would make all our edges probabilistic. But we leave that for future work. Using this notion of edge weights, we define $EdgeScore(P)$ of a path $P$ as product of all the edge weights in it. For $P$ to be a good explanation, its $EdgeScore$ should be high.

Next, we define an operator, $RelComposition(P)$, which performs a composition of all the relations in the path $P$. It returns a vector in the same space as the relation embeddings learnt by the model. We follow previous work to use a Hadamard product of the embeddings of all relations in the path [Guu et al., 2015]. For example, for $P(s, o) = r_1(s, u_1), r_2(u_1, o)$, $RelComposition(P) = r_1 \bullet r_2$.

Now, let the model $M$ predict an entity $o$ for the query $r(s, ?)$. To quantify the plausibility of an explanation path $P(s, o)$ for the prediction $r(s, o)$, we define a plausibility score as:

$$Plausibility(P(s, o), r(s, o)) = sim_M \left( RelComposition \left( P(s, o) \right), r \right) \cdot EdgeScore(P) \quad (1)$$

In the above equation, $EdgeScore(P)$ captures all the entity similarities in the path $P$ and $sim_M(RelComposition(P), r)$ captures whether composition of relations in the path entail the given relation $r$ or not. $sim_M$ function is as defined earlier.

Task of OxKBC is to select the most plausible explanation path. However, one can easily observe that $Plausibility$ score might not be comparable for two paths of unequal lengths. Even for two equal length paths, it could be difficult to compare due to different edge types. For example, consider two paths $P_1$ and $P_2$ with length two but different edge types: $P_1(s, o) = \sim_M (s, u_1)$, $r_1(u_1, o)$ and $P_2(s, o) = r_1(s, v_1), r_2(v_1, o)$. $P_1$ has a similarity edge followed by a relation edge

and $P_2$ has two relation edges. Clearly $Plausibility(P_1, r(s, o)) = sim_M(s, u_1) \cdot sim_M(r_1, r)$ and $Plausibility(P_2, s, r(s, o)) = sim_M(RelComposition(r_1, r_2), r)$ are not comparable as former involves product of two similarities whereas the latter has only one term. To resolve such discrepancies, we define the notion of a second–order template which aggregates comparable explanation paths. Once we have defined a template, then OxKBC would be left with the job of selecting the most appropriate template and ground it to generate an explanation.

### 3.2 Templates: Aggregation of similar Explanation Paths

Depending on the sequence of edge types in it, we classify an explanation path into a second–order template in which we quantify over both relations and entities. Each template corresponds to a fixed sequence of edge types. This aggregation is required because the entire space of explanation paths is huge and it is not feasible to compare all of them. Hence, to operationalize the selection process, we aggregate them into templates. We use these templates to explain the prediction $o$ for a query $r(s, ?)$:

---

T1. **Relation Similarity**: $r'(s, o) \in KB$ and $r \sim_M r'$
T2. **Entity Similarity**: $r(s', o) \in KB$ and $s \sim_M s'$
T3. **Entity & Relation Similarity**: $s \sim_M s'$ and $r'(s', o) \in KB$ and $r \sim_M r', r \neq r'$
T4. **Two Length Relation Similarity**: $r_1(s, u_1) \in KB$ and $r_2(u_1, o) \in KB$ and $r \sim_M RelComposition(r_1, r_2)$

---

Preliminary investigation of potential explanations for our datasets reveal that these templates are able to explain most of the predictions. Our work naturally extends to other templates, such as "Entity & two-length Relation Similarity", but we do not implement them for our datasets.

We note that while other works have considered rules based on relation similarity earlier, e.g. [Yang et al., 2015] mine first order one length and two length rules (T1 and T4 respectively), we are the first one to use the notion of entity similarity for generating explanations. Intuitively, entity similarity captures an aggregation of many 2 length paths: a high similarity score between $e$ and $e'$ implies that they can be used interchangeably, i.e. $\exists r \in R$ and $u \in E$ s.t. both $r(e, u), r(e', u) \in KB$, i.e. $P(e, e') = r(e, u), s(u, e')$ is a path in KB if $s = r^{-1}$ is also in the set of relations $R$, and this holds for many choices of $r$ and $u$, implying existence of many such paths. While rule mining may discover just one path between $e$ and $e'$, entity similarity may hop directly using an entity similarity edge, e.g. for second fact in Table 3.

We also note that since our templates are second order, they have the representation power to subsume first order rules. For example, T1, which infers that $r(s, o)$ because $r'(s, o) \in KB$ and "$r$ is similar to $r'$" subsumes one length rule "$r' \implies r$" as learnt in many other rule mining systems.

While annotating the data, we observed that in few cases $o$ is a prediction for a given query $r(s, ?)$ because in the $KB$ it is the most frequent entity for the relation $r$. For example, most of the websites ($s$) in the $KB$ are in 'Language' ($r$) 'English' ($o$), and hence model learns to predicts 'English' whenever asked for 'Language' of any website. To account for such explanations, we consider a frequency based template as well:

---

T5. **Frequent for Relation**: $o$ is a highly frequent tail entity for the relation $r$ in $KB$

---

These templates may not be able to explain every prediction, hence, we also introduce a default template (T0) that corresponds to *'no explanation'*. Table 3 lists sample explanations (generated by OxKBC) on FB15K test data along with template IDs.

**Template Scoring:** The *Compatibility* function between a path $P$ and a prediction $r(s, o)$ as defined in equation 1 offers a natural way of scoring an explanation path $P$ for the given prediction. Since a given similarity based template represents comparable paths with same sequence of edge types in it, we can quantify it with the explanation path having maximum score. Accordingly, we define template score, $T_i^S$, for a similarity based template $Ti$ as:

$$T_i^S(s, r, o) = \max_{P \in Ti} Compatibility(P, r(s, o)) \quad \forall i \in \{1, \ldots, 4\} \tag{2}$$

Inspired by the notion of selecting the best amongst all paths in a given similarity based template, we define score of frequency based templates T5 as the normalized frequency of the corresponding relation, tail entity pair $(r, o)$ in $KB$:

$$T_5^S(s, r, o) = \frac{|\{s' : r(s', o) \in KB\}|}{|\{(s', o') : s', r(s', o') \in KB\}|} \tag{3}$$

These goodness scores act as important features in the template selection module (section 3.3). Once a template is selected, an explanation in English is produced as shown in Table 3. For templates T1-T4, we ground them with the entities/relations in the explanation path corresponding to the $argmax$ of the $Compatibility$ score.

**Faithfulness of Explanation Paths to the TF Model:** Multiplicative models like TypeDistMult use three-way products to compute scores of a triple. So, a high dot product (and therefore, cosine score) between two relation embeddings $r$ and $r'$ (or two entity embeddings) fundamentally represents that model $M$ considers the two relations (entities) as somewhat replaceable. This idea is exploited in our similarity templates. Consider, as an example, a query $r(s, ?)$ and the corresponding prediction, $o$, for the tail entity by the model $M$. Say OxKBC explains the prediction using T1 by finding the best fact $r'(s, o)$ in the KB such that $r \sim_M r'$ (cosine score between the two is high). Now, since $r'(s, o)$ is in the KB, model score for $r'(s, o)$ must be trained to be high, and hence, Hadamard product of the embeddings of $s$ and $o$ is likely to be closely aligned with the embedding of $r'$. Because $r \sim_M r'$ Hadamard product of the embeddings of $s$ and $o$ is also closely aligned with that of $r$, resulting in a high model score for $r(s, o)$. Thus, this explanation is one reasonable clue why model $M$ may have decided to make this prediction in the first place. Similar arguments can be made for T2 T3 and T4.

### 3.3 Selection Module

OxKBC's selection module (SM) decides which template to select for explaining a given prediction. It uses a 2 layer MLP for this task. For each template $i$, it takes an input feature vector and outputs a score $S_i^{\text{SM}}$, representing SM's belief that this template is a good explanation for the prediction. The scores $S_i^{\text{SM}}$ are converted into probabilities $P_i^{\text{SM}}$ through a $softmax$ layer. OxKBC chooses the template with the highest probability.

**Input Features:** For a given query $r(s, ?)$ and a template $i$, we compute template goodness scores $T_i^S(s, r, u) \forall u \in E$. This defines a distribution of goodness scores over tail entities for a given input query $r(s, ?)$. To explain a prediction $r(s, o)$, we construct a feature vector such that it captures the relative score $T_i^S(s, r, o)$ of $o$ w.r.t the distribution $T_i^S(s, r, u)$ over $u \in E$. We claim that this feature vector has all the information to decide if this template is good for explaining the given prediction or not. It maintains the distribution level global features w.r.t. the query $r(s, ?)$: max, mean, std deviation of the distribution, and prediction specific features: score, rank and other statistics of

the specific prediction $o$. See Appendix A.3 for exact definition of features. Since we do not have goodness scores for T0, SM considers its features as trainable parameters.

**Training:** Manual labeling of explanations is time-consuming, so we primarily train SM in unsupervised[3] and semi-supervised settings. For completeness, we also report on a fully supervised setting on our small annotated dataset. For the unsupervised setting, the key insight is that for most triples known to be true, we may expect some explanation, but we do not know which template to select. However, for a randomly selected negative sample, the selection module must always select T0 as we do not expect any explanation for it. In the absence of template level annotation, we rely on distant supervision from this weak signal. SM formulates it in a multi-instance learning (MIL) framework [Dietterich et al., 1997] (see Appendix A.4 for details), where feature vector for each template is an instance, and supervision is at the bag level (positive or negative labels on a triple).

**Loss Function:** We operationalize this through an unsupervised loss function. When classifying a positive triple, SM gains reward ($R_{pos}$) proportional to the highest template probability amongst templates other than T0. To account for the case that sometimes no template might be available even for a correct triple, SM gets a lower reward even for selecting T0 (eq. (4)). For negative samples, the module is penalized ($R_{neg}$) if it selects any template other than T0 – it gets a negative reward proportional to total probability of all the explanation templates other than T0, and positive reward proportional to probability of T0 (eq. (5)).

$$R_{pos} = \rho_{pos} p_0^{\text{SM}} + \max_{i \neq 0} p_i^{\text{SM}} \qquad (4)$$

$$\mathcal{L}_{\text{unsup}} = -R_{pos} - R_{neg} \qquad (6)$$

$$R_{neg} = p_0^{\text{SM}} - \rho_{neg} \sum_{i \neq 0} p_i^{\text{SM}} \qquad (5)$$

$$\mathcal{L}_{\text{semi}} = \mathcal{L}_{\text{unsup}} + \lambda_1 \mathcal{L}_{\text{sup}} + \lambda_2 \mathcal{L}_{\text{PR}} \qquad (7)$$

In semi-supervised setting, we assume access to a small train set of labeled explanations, i.e., the correct template for a fact is annotated. We add a supervised loss term ($\mathcal{L}_{\text{sup}}$), which is the standard cross-entropy loss over the set of labeled examples. Additionally, we project the template label distribution from labeled set into unlabeled data through posterior regularization [Ganchev et al., 2010]. We add a KL divergence loss ($\mathcal{L}_{\text{PR}}$) between labeled probability distribution over templates in train set and learned distribution. All hyperparameters $\rho$s and $\lambda$s are tuned on a small devset.

## 4. Experiments

The goal of our experiments is to answer the following research questions: (1) What is the benefit of adding a small amount of supervision on the overall performance of our model? (2) How good are our explanations compared to existing approaches? (3) Are our explanations faithful to the underlying $TF$ model? To answer the first question, we perform an internal evaluation using the annotations made by the authors. To answer the second question, we perform a human evaluation using workers on Mechanical Turk. To answer the third question, we design an experiment to observe the impact of removing a fact from the KB which is part of an explanation on the quality of prediction made by the $TF$ model. Next, we describe our experiments in detail.

**Datasets:** Our experiments use the standard KBC datasets, FB15K [Bordes et al., 2013] and YAGO3-10 [Dettmers et al., 2018]. FB15K is a subset of the original Freebase [Bollacker et al., 2008] and has $592,213$ facts with $14,951$ entities and $1,345$ relations. A large fraction of this knowledge base describes facts about actors, awards, movies and sports. YAGO3-10 is a subset of YAGO [Suchanek

---

3. We use a very small set of 20 labeled samples as dev set for hyperparameter tuning.

et al., 2007] and has $1,089,040$ facts with $123,128$ entities and $37$ relations. Most of the facts describe attributes like citizenship, gender, profession, etc. for various people.

**Models:** We use TypedDistMult [Jain et al., 2018] as our underlying tensor factorization model $M$, since its code is freely available and it has near state-of-the-art performance on these datasets. To train it, we retain the exact train, dev and test folds used in previous work.

**Training Data:** For training SM, we randomly select 100 samples from the original dev set, where $M$ makes correct prediction at the top. Two authors annotate each sample with the template producing the best explanation with inter-annotator agreement (Cohen's Kappa) of $0.87$, and resolve conflicts through discussion. This labeling took about 3 hours. Out of 100 training samples, 20 samples is used as dev set for hyper parameter tuning. Remaining 80 samples are used as train set in semi-supervised and supervised setting to compute supervised loss ($\mathcal{L}_{\text{sup}}$). The entire train data of the original model $M$ is used for computing the unsupervised loss ($\mathcal{L}_{\text{unsup}}$) in semi-supervised and unsupervised setting.

**Templates:** We evaluate only those templates via $SM$ that are chosen at least once in our manual annotation. For FB15K, this includes T1, T2, T4 and T5 and for YAGO, only T1, T2 and T5.

## 4.1 Internal Evaluation

Similar to labelled training data, we annotate around 135 randomly sampled facts from the original test data with the template producing the best explanation. We note that in our internal evaluation, we use only those test queries where the true fact was scored at the top by our model since it is not quite clear how to evaluate the explanation of an incorrect tuple. We report micro-F1 for these templates over this unseen labelled test data as our evaluation metric. We repeat each experiment three times by changing random seeds and report the mean and standard deviation (See Appendix A.5 for details on hyperparameters, training time and computing infrastructure).

Unsupervised training is done using only the weak supervision on the unlabeled train data (equation 6, whereas, the supervised training is done using only 80 labeled data points. Individually, both may not achieve as high

| Dataset | Unsupervised | Supervised | Semi-supervised |
|---------|--------------|------------|-----------------|
| **FB15K** | $0.537 \pm 0.012$ | $0.421 \pm 0.089$ | $0.641 \pm 0.011$ |
| **YAGO** | $0.699 \pm 0.384$ | $0.965 \pm 0.010$ | $0.968 \pm 0.004$ |

Table 1: Micro-F1 score of OxKBC (mean $\pm$ std deviation)

performance as our semi-supervised setting, which uses the entire training data as unlabeled data along with manually annotated 80 labeled data points. It also includes additional evidence using the KL divergence term that is computed over unlabeled data.

Table 1 reports the results from this evaluation. We observe that semi-supervised learning indeed outperforms both other settings. We also note that unsupervised setting can have a huge variance, such as for the YAGO dataset. This is not surprising, since it can be very difficult to train the model using only distant supervision in the form of unsupervised loss term. As a result, in a couple of random runs, it learns a degenerate model where all predictions collapse randomly to any one of the templates. Adding KL divergence term along with a small amount of labeled data (80 samples) in the overall loss term helps in maintaining the same posterior distribution over the output space and ensures that it does not learn a degenerate model, thus reducing variance for semi-supervised setting.

The scores on YAGO are particularly high, because there are very few relations for which $M$ makes correct predictions at the top, making the role of OxKBC easy. Overall, our results are in line with our expectation that semi-supervised learning will outperform purely supervised or unsupervised learning, by exploiting statistical regularities present in large amounts of unlabeled data.

## 4.2 User Evaluation

Since explanations are ultimately for end users, we compare the quality of our explanations over other proposed approaches, by a user study over Amazon Mechanical Turk[4] on FB15K dataset.

**Baseline**: For implementing rule mining baseline, we adapt the procedure as defined in [Yang et al., 2015] to mine closed-path logical rules upto 3 length and use embeddings trained by $M$ to prune the exponential search space. The rules are ordered by confidence, defined as the ratio of number of correct predictions using the rule and the total number of predictions possible using body of the rule. Refer to section 5 in [Yang et al., 2015] for more details. In the first version of our experiments, we noticed that for many predictions, rules did not provide any explanation, which led to workers preferring OxKBC almost always over rule mining. To fix this, we reduced the thresholds, but that resulted in selection of longer rules much more frequently because of their rarity (making their confidence high). Experiments with this baseline resulted in workers preferring OxKBC over rules 84% of the time. But for a given prediction, we notice that a rule of length 1 is almost always more convincing than a rule of length 2 and so on. Hence we modify the rule selection procedure to prefer a lower length rule whenever present. Within same length, rule with highest confidence in selected. We report our results using this strong version of the baseline.

### 4.2.1 QUALITY OF EXPLANATIONS

We ask workers to compare explanations generated by OxKBC with explanations derived from the baseline. Both the systems use the same underlying base model $M$ to generate their explanations. We randomly select 450 different facts from the test data and generate explanations from the two explanation engines. Out of these, 97 have the exact same explanation from both engines, and 21 have no explanation from both engines. For the remaining, we ask five different workers to select the better explanation. They are also given an option to reject or accept both explanations. In cases, where one engine offers no explanation, we ask workers whether the other explanation is better or worse than not having an explanation. For explaining entity similarity, we also generate a context for why two entities are similar and the user can hover to see the facts where the entities are interchangeable to see the rationale for why $M$ considers two entities similar. Finally, we use majority voting to decide which explanation is better for a fact.

|  | OxKBC Better | Rules Better | Tie | Total |
|---|---|---|---|---|
| **Overall** | **145** | 49 | 18 | 212 |
| **Both** | **118** | 40 | 18 | 176 |
| **OxKBC** | **2** | 2 | 0 | 4 |
| **Rules** | 25 | **7** | 0 | 32 |

(a) Comparison between explanations for GT@1

|  | OxKBC Better | Rules Better | Tie | Total |
|---|---|---|---|---|
| **Overall** | **94** | 16 | 10 | 120 |
| **Both** | **76** | 10 | 10 | 96 |
| **OxKBC** | **0** | 3 | 0 | 3 |
| **Rules** | 18 | **3** | 0 | 21 |

(b) Comparison of explanations for N-GT@1

Table 2: OxKBC vs. rule mining under different scenarios: **Overall:** when either of the two systems give an explanation; **Both:** When both give an explanation; **OxKBC:** When only OxKBC gives an explanation (explained above); Similarly for 'Rules'. When only one of the systems produces an explanation (last two rows), we ask workers whether it is better or worse than having no explanation. If it is selected as being worse, we mark the other system, producing 'no explanation', as better.

---

4. **Selecting Workers**: See Appendix B for details on how we select quality workers for our experiments.

We present the results in Table 2 and split the analysis into two parts: one when model's top prediction matches the gold annotation in the data, denoted as GT@1 (Gold Truth@1) in Table 2a, and the other when it doesn't match the gold: N-GT@1 (Not–Gold Truth@1) in Table 2b. We see that not only are the explanations from OXKBC preferred over rule mining for GT@1, but even for the predictions for samples in N-GT@1, OXKBC gives a more probable explanation than rule mining. Note that the workers are never told if the given fact is true or not. Table 3 lists some facts, where OXKBC outperformed the explanations generated by rule mining and vice-versa.

| Fact | Explanation from OXKBC | Explanation from Rule Mining |
|---|---|---|
| (Academy Award for Best Sound Mixing, has nomination for, WarGames) | [T4] (Academy Award for Best Sound Mixing, has nominee, Willie D. Burton) and (Willie D. Burton, was an award nominee for, WarGames) | (Academy Award for Best Sound Mixing, has nomination for, On Golden Pond) and (On Golden Pond, was nominated for, Academy Award for Best Cinematography) and (Academy Award for Best Cinematography, has nomination for, WarGames) |
| (The Last King of Scotland, has genre, Drama) | [T2] (The Lives of Others, has genre, Drama) and "The Last King of Scotland" is similar to "The Lives of Others" | (The Last King of Scotland, has actor, Forest Whitaker) and (Forest Whitaker, won an award for, Bird) and (Bird, has genre, Drama) |
| (47th Annual Grammy Awards, had an award category, Grammy Awards for Song of the Year) | [T2] (50th Annual Grammy Awards, had an award category, Grammy Awards for Song of the Year) and "47th Annual Grammy Awards" is similar to "50th Annual Grammy Awards" | (47th Annual Grammy Awards , is an instance of repeating event:, Grammy Awards) and ( Grammy Awards, category , Grammy Awards for Song of the Year) |
| (Actor, is the profession of, John Lithgow) | [T2] (Musician, is the profession of, John Lithgow) and "Actor" is similar to "Musician" | (Actor, is the profession of, Henry Winkler) and (Henry Winkler, has profession, Writer) and (Writer, is the profession of, John Lithgow) |

Table 3: OXKBC was preferred to Rule Mining for first two facts, rule mining won in third fact, and none of the explanations were accepted in the last example.

| Fact | Explanation from OXKBC |
|---|---|
| (Sound Mixer, had some job associated with film, The Motorcycle Diaries) | (Supervising Sound Editor, had some job associated with film, The Motorcycle Diaries) and ("Sound Mixer" is similar to "Supervising Sound Editor") |
| (Composer, is the profession of, Danny Elfman) | (Composer, is the profession of, John Kander) and (John Kander, is an award co-nominee with, Danny Elfman) |
| (2010 Winter Olympics, has a participating country, Czech Republic) | (2002 Winter Olympics, has a participating country, Czech Republic) and ("2010 Winter Olympics" is similar to "2002 Winter Olympics") |

Table 4: OXKBC explanations from N-GT@1 when the top prediction is correct in real world.

### 4.2.2 QUALITATIVE ANALYSIS OF EXPLANATIONS FOR INCORRECT PREDICTIONS

From Table 2b, we observe that workers prefer OXKBC's explanations even when the prediction of the model $M$ is not marked as correct (N-GT@1). We randomly select 100 samples from N-GT@1 and manually analyze the explanations generated by OXKBC. For 77 out of 100 samples, OXKBC generated convincing explanations. On further investigation, we discovered that 52 out of 77 predictions were indeed correct in real-world but the model was penalized because the fact was

not present in the KB or dev/test sets. This is an issue of the incompleteness of KB and demonstrates that good explanation can help in augmenting the KB with more facts (examples in Table 4). In the second case, we looked at remaining 25 odd examples where top model prediction was actually incorrect (in real-world). For 15 out of 25 samples, explanations looked reasonable even though the facts were not true. Table 5 lists few examples. While statistically a good prediction, in this particular case, the facts were untrue. We consider that these errors should not be used for model improvement, since the explanations are reasonably convincing even if the facts are untrue. The remaining 10 examples were the cases where the model made genuine errors (e.g., type related errors) and we did not get any proper explanation. These are the cases where we can potentially improve the model, e.g. by reranking until we get a plausible explanation. Doing this experimentation further as part of a controlled user study is a direction for future work.

| Fact | Explanation from OxKBC |
|---|---|
| (Sex and the City, has actor, Fergie) | (Sex and the City, is an award nominating work of, Fergie) |
| (A Fish Called Wanda, was nominated for Academy Award for, Best Picture) | (A Fish Called Wanda, is an award nominating work of, Michael Shamberg) and (Michael Shamberg, was nominated for, Academy Award for Best Picture) |
| (Bobby McFerrin, music artist has genre, Rock music) | (Billy Joel, music artist has genre, Rock music) and ("Bobby McFerrin" is similar to "Billy Joel") |

Table 5: OxKBC explanations for samples from N-GT@1 when the predictions are actually incorrect.

### 4.3 Faithfulness of Explanations

In these set of experiments, we set out to ask if the explanations generated by OxKBC are faithful to the underlying $TF$ model or not? We note that explanations consist of supporting facts, e.g., for a prediction $r(s, o)$, if OxKBC produces an explanation using template T1, i.e., $r(s, o)$ because $r'(s, o)$ and $r' \sim r$, then $r'(s, o)$ is the supporting fact from KB used for grounding the template. We ask how important are the supporting facts in underlying $TF$ model's decision making? Specifically, we observe the change in the quality of predictions caused by removing

|  | MRR | | | Control MRR | | |
|---|---|---|---|---|---|---|
| **Steps** | **T1** | **T2** | **T4** | **T1** | **T2** | **T4** |
| 1 | 1.00 | 1.00 | 1.00 | 1.00 | 1.00 | 1.00 |
| 1 | 0.50 | 0.60 | 0.40 | 0.94 | 0.95 | 0.96 |
| 2 | 0.30 | 0.34 | 0.24 | 0.88 | 0.89 | 0.89 |
| 5 | 0.20 | 0.18 | 0.13 | 0.69 | 0.57 | 0.63 |

Table 6: MRR over 100 samples when the probability of the supporting fact in the explanation is reduced. Control MRR reports the MRR when probability of any other related fact in the KB is reduced.

such supporting facts from the KB, compared to the change observed when a random fact connected with the prediction (control) is removed. Greater the resulting change, higher our confidence that the model used supporting facts as evidence for the prediction, and that our explanations are indeed faithful.

To achieve the above objective, we perform the following experiment. For each template, we randomly select 100 correct predictions for which OxKBC outputs a grounding of that template as an explanation. We then take a few gradient steps to reduce the probability of the supporting fact(s) selected by OxKBC, i.e. $r'(s, o)$ in the above example, and compute the rank of the entity $o$ for query $r(s, ?)$. As a control experiment, for each prediction $r(s, o)$, we randomly select a fact from KB changing either the subject $s$, object $o$ or the relation $r$ from the triple and take a few gradient steps to reduce the probability of thus selected fact. We again compute the rank of the entity $o$

for query $r(s, ?)$. We report the MRR for the two settings (non-control vs. control) over the $100$ randomly selected predictions for each template in Table 6.

Our results show that there is a reduction in 'Control MRR' as well. This is expected as the facts selected in the control experiment are not completely random but in the neighborhood of the prediction and may have some impact on the model score for the prediction. However, as can be seen from the table, reducing the model's belief for the supporting fact results in a much sharper drop in its confidence of the correct prediction (as shown by sharp decrease in MRR vs control MRR). In other words, if the supporting facts were not correct, then model would not have ranked the object $o$ high enough for the query $r(s, ?)$. This validates the hypothesis that explanations generated by OxKBC are indeed faithful to underlying $TF$ model.

## 5. Discussion

OxKBC currently only outputs one best explanation: the top grounding of the selected template. To generate alternative explanations, it can always generate the next best groundings of the selected template. However, alternative good explanations may also come from a different template. This will require us to learn a Selection Module that can compare the second best grounding of a top template with the best grounding of the second best template (and so on). Given that our design of OxKBC assumes minimal available supervision, more thought is needed on how to compare these.

**Sub–graph as an explanation:** We also note that while explanations are often paths in a KG, sometimes they may be not, and we may need to generalize the notion of explanation as a "path" to explanation as a "sub–graph". We believe that our general approach may be extensible to subgraph explanations. We can extend the explanation grammar to allow for subgraph rules. We can then define new templates and provide a compatibility function for the template. The Selection Module should follow as is. For instance, $isSpouseOf(X, Y)$ because $isFatherOf(X, Z)$ and $isMotherOf(Y, Z)$ could be an instance of some template where $P \leftarrow P_1, P_2 \ s.t. \ Head(P_1) = Head(P), Head(P_2) = Tail(P)$ and $Tail(P_1) = Tail(P_2)$. We did not study this in depth because, in our preliminary analysis on our datasets, paths came out as natural explanations, and more generic subgraphs were not found essential. We leave these two directions for future research.

## 6. Conclusions

We present OxKBC, a novel approach for providing post-hoc explanations for tensor factorization based KBC models. We introduce the idea of entity similarity edges in the knowledge graph and use paths in the augmented knowledge graph as explanations. We quantify over both relations and entities to define second–order templates for aggregating explanation paths and train a neural selection module to pick the best template for a given prediction. The neural model can be trained in a minimally supervised fashion due to a novel loss function. Based on a user study done on mechanical turk, we find that explanations generated by our system are significantly better than those generated from rule mining. We perform additional experiments showing that the explanations generated by OxKBC are faithful to the predictions of the underlying model. In the future, we hope to extend the explanation engine for debugging the KBC model and improving its performance. We make our code publicly available for research at https://github.com/dair-iitd/OxKBC.

## Acknowledgement

We thank IIT Delhi HPC facility[5] for computational resources. We thank Sameer Singh for useful discussions on the work. Mausam is supported by grants from Google, Bloomberg, 1MG and Jai Gupta chair fellowship by IIT Delhi. Parag Singla is supported by the DARPA Explainable Artificial Intelligence (XAI) Program with number N66001-17-2-4032. Both Mausam and Parag Singla are supported by the Visvesvaraya Young Faculty Fellowships by Govt. of India and IBM SUR awards. Any opinions, findings, conclusions or recommendations expressed in this paper are those of the authors and do not necessarily reflect the views or official policies, either expressed or implied, of the funding agencies.

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

## Appendix A.

### A.1 Comparison of TF based KBC Models with Path based Approaches

Methods like DeepPath, MINERVA, are inherently explainable as they work directly with paths, unlike TF based methods. However, such methods either do not scale or do not perform as well as TF based methods for large scale KBs such as FB15k. MINERVA is the latest of the two techniques and table 5 in [Das et al., 2018] shows it to be better than DeepPath and hence in the following we

compare against only MINERVA. On FB15k-237, which is a smaller version of FB15k with 237 relations, MINERVA reports HITS@1 of 0.217 (table 4 in [Das et al., 2018]) whereas our underlying model Type-Distmult reports significantly higher HITS@1 of 0.293 (table 3 in [Jain et al., 2018]). [Das et al., 2018] doesn't report performance of MINERVA on full FB15K, though their code is publicly available. In our preliminary experiments, we trained their code on FB15K using the same hyperparameters as that for FB15K-237 provided by the authors and we achieved HITS@1 of only 0.25 and upon optimizing it further, we could reach only 0.29. On the other hand, [Jain et al., 2018] reports HITS@1 of 0.66 for the Type-Distmult TF model used in our experiments.

### A.2 Pedagogical Approach

The approach as mentioned in [Gusmão et al., 2018] does not scale to the size of FB15K dataset. They perform their experiments on a toy FB13 dataset where they use only 13 relations of the entire FB15K dataset. We tried their methodology on FB15K but memory and storage requirements where intractable. According to our calculations, it would take 25 TB of storage and 20 days of compute, using the available code on an available system with 256 GB of RAM and 40 Intel(R) Xeon(R) CPU @ 2.80GHz cores.

### A.3 Template Features

Table 7 enumerates the template features used as input to the Selection Module.

| Global Features for a query $r(s,?)$ | | Prediction specific features for $r(s,o)$ | |
|---|---|---|---|
| max score | $\max\limits_{u \in E} T_i^S(s,r,u)$ | score | $T_i^S(s,r,o)$ |
| distr. mean | $\operatorname*{mean}\limits_{u \in E} T_i^S(s,r,u)$ | similarity | $sim_M\left(o, \arg\max\limits_{u \in E} T_i^S(s,r,u)\right)$ |
| distr. std | $\operatorname*{std}\limits_{u \in E} T_i^S(s,r,u)$ | rank | $\operatorname*{rank_o}\limits_{u \in E} T_i^S(s,r,u)$ |

Table 7: Input Features for a prediction $r(s,o)$ from $i^{th}$ template $Ti$

### A.4 Architecture of Selection Module

We use a two layer MLP with 90 and 40 neurons in two hidden layers as our model for the selection module. We imitate the Multi-Instance Learning setting, where the same model is used to output scores for each template $Ti$. Finally, we take a softmax over all these scores, to return a probability distribution over the space of templates, and select the one with maximum probability as our best explanation template. Figure 1 shows the overall architecture.

### A.5 Experiment Details

**Hyperparameters:** We use SGD optimizer, with a learning rate of $0.001$, momentum of $0.9$ and a batch size of $2048$ for all our experiments. Table 8 states the hyper parameters used for each setting.

**Computing Infrastructure and Time Analysis**: For generating feature vectors corresponding to each template, we used a machine with 16 Intel(R) Xeon(R) W-2145 CPU @ 3.70GHz cores and 128 GB of RAM. No GPU memory was required. It required about 5 hours to preprocess and generate

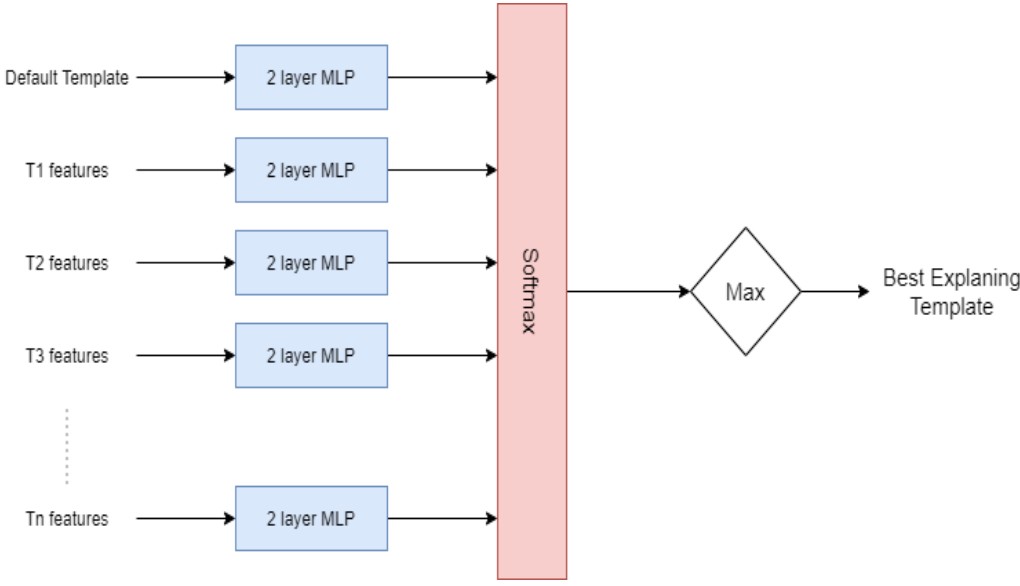

Figure 1: Architecture of our Selection Module. The underlying two-layer MLP is same when predicting score for every template.

| Dataset | Setting | $\rho_{neg}$ | $\rho_{pos}$ | $\lambda_2$ | $\lambda_1$ |
|---------|---------|--------------|--------------|-------------|-------------|
| FB15K | Un-supervised | 1 | 0.1 | 0 | 0 |
| | Semi supervised | 1 | 0.125 | 1 | 1 |
| YAGO | Un-supervised | 0.5 | 0.2 | 0 | 0 |
| | Semi supervised | 0.25 | 0.25 | 1 | 1 |

Table 8: Hyper-parameters used for various settings

the feature vectors for the train and dev sets. For training the two layer MLP in SM, we used a GPU with 16 GB memory, though the GPU memory required is less than 1 GB (for a batch size of 2048). The module gets trained in about 20 minutes.

At test time, to explain a prediction in OXKBC most of the time is spent in extracting features for different templates. For FB15K, it takes on average around 500 milliseconds per test sample to extract template features using a single core of an Intel(R) Xeon(R) W-2145 CPU @ 3.70GHz. Once features are extracted, it takes only 0.38 milliseconds to select the best template using the selection module. On the other hand, in rule mining, once the rules are mined, it takes only 2.04 milliseconds to select the best rule for a given test sample. Even though at test time, OXKBC is much slower than rule mining, we believe it is still fast enough to be usable, especially for a human interacting with an explanation interface. We also note that this extra inference time comes with the advantage of improved quality of explanations.

## Appendix B. Mechanical Turk Experiment Details

**Selecting Workers**: To make sure that the workers really understand the task, we create an interactive tutorial of 6 questions in which users can select different options and get feedback about each of them. Figure 2 shows a screenshot from the tutorial. To make our tutorial unbiased, we select questions which expose the good and bad sides of both the systems, and at no point are the workers told which explanation comes from which system. On completing the tutorial, they perform the task of comparing the two explanations on a small set of 10 gold questions. For each question, they are also asked to give a subjective reason for their choice. We analyze their answers quantitatively and their reasons qualitatively and manually cherry-pick the diligent and reasonable workers. All subsequent experiments are done with these quality workers.

**Human Intelligence Task:** Figure 2 shows a screenshot of the task given to workers. We ask them to chose which out of the two explanations are better. They can select or reject both the explanations as well. In case only one system gives an explanation, workers choose if it is better than no explanation or not. In order to explain the task better, we create an interactive tutorial where workers can see why some explanation is inferior compared to its counterpart. For explaining entity similarity or high frequency of entity, we also generate a context of why two entities are similar or why is some entity frequent and the user can hover over bordered boxes to see the additional facts.

(a) Instructions provided to workers

**Question:** **Desperate Housewives** **was nominated for** _____ ?
**Answer:** **Primetime Emmy Award for Outstanding Comedy Series**
**Explanation A:** **Desperate Housewives** **is an award nominating work of** **Marc Cherry** and **Marc Cherry** **was nominated for** **Primetime Emmy Award for Outstanding Comedy Series**
**Explanation B:** **Desperate Housewives** **has regular appearance of** **Lyndsy Fonseca** and **Lyndsy Fonseca** **has regular appearance in** **How I Met Your Mother** and **How I Met Your Mother** **was nominated for** **Primetime Emmy Award for Outstanding Comedy Series**

○ A is better than B
◉ B is better than A
○ Both A and B are equally good
○ Both A and B are bad

**Wrong.**
**Just because another show was nominated for an award and both have regular appearance of a common actor (Lyndsy Fonseca), it doesn't indicate strongly that 'Desperate Housewives' should also be nominated for the same award.**
**Moreover, Explanation A not only gives an explanation, it also tells the writer / producer of Desperate Housewives (Marc Cherry) who was nominated.**

(b) The tutorial has 6 of such questions which we use to tell workers why some explanations are better than others.

**Question:** **Turun Palloseura** **football team has a play position** _____ ?
**Answer:** **Midfielder**
**Explanation A:** **Turun Palloseura** **football team has a play position** **Defender** and **Defender** **is a position in the football team** **Lierse S.K.** and **Lierse S.K.** **has a position of** **Midfielder**
**Explanation B:** In our Knowledge Base, *many football teams (598 of 745)* have players at Midfielder
(Australia national soccer team , Olympique de Marseille and *596 more...* ) football team has a play position Midfielder

○ A is better than B
○ B is better than A
○ Both A and B are equally good
○ Both A and B are bad

(c) Example query with frequent entity context

Figure 2: Human Intelligence Task on Amazon Mechanical Turk

