# OpenReview forum: "OxKBC: Outcome Explanation for Factorization Based Knowledge Base Completion"
_AKBC.ws/2020/Conference — AKBC 2020_

### Official Review · AnonReviewer1 · 2020-03-06
**Good paper but needs more experimentation**

**Rating:** 5
**Confidence:** 4

**Review:**


Clarity: The paper is easy to read and well written

Originality: To the best of my knowledge, this is a novel work. Most of the related relevant works are cited by the authors.

Significance: In this paper authors propose a mechanism to generate explanations for the inferred KBC facts. This perfectly aligns with the growing efforts on explainable AI systems. I believe that this work will be relevant to the conference audience.

Quality: The overall quality is represented by my rating

Details:
In this paper, authors propose a method to generate explanations for the inferred facts for KBC task. They compare their technique with an existing rule-miner technique and show that their system produces more intuitive explanations.

I have following comments/questions:
1. Authors mention systems that use RL for the same task - DeepPath, MINERVA. However they don't report any quantitative comparison.
2. The proposed method explains outcome of the system (i.e. it is outcome explanation). As per authors this method tend to be a bit slower compared to model explanation techniques. Some quantitative comparison would be useful.
3. In Table 1 for Unsupervised setting experiment with YAGO, authors report a standard deviation of 0.384 - is this correct? If it is then it's very high.
4. In the same table for FB15K the difference in performance of Semi-supervised vs Supervised setting is very high. What's the reason?
5. It would be good to also see change in the performance as the amount of supervision changes in the semi-supervised setting. At the moment authors have reported only for one setting.

---

> ### Author Response · Authors · 2020-04-06
> **Reply to AnnonReviewer1**
>
> Dear reviewer,
> Thanks for your valuable comments and suggestions. Below we have tried our best to address each of your concerns in the same order as in the review:
>
> 1. As mentioned in “Related Work” section (#2), 1st para, 3rd line, methods like DeepPath, MINERVA, are inherently explainable as they work directly with paths, unlike TF based methods.  However, such methods either do not scale or do not perform as well as TF based methods for large scale KBs such as FB15k. MINERVA is the latest of the two techniques and table 5 in [1] shows it to be better than DeepPath and hence in the following we compare against only MINERVA.  On FB15k-237, which is a smaller version of FB15k with only 237 relations, MINERVA reports HITS@1 of 0.217 (table 4 in [1]) whereas the underlying model used in our paper (type-Distmult) reports significantly higher HITS@1 of 0.293 (table 3 in [2]). [1] doesn’t report the performance of MINERVA on full FB15k, though their code is publicly available. In our preliminary experiments, we trained their code on FB15k using the same hyperparameters as that for fb15k-237 provided by the authors we achieved HITS@1 of only 0.25 and upon optimizing it further, we could reach only 0.29. On the other hand, [2] reports HITS@1 of 0.66 for the type-Distmult TF model used in our experiments.
>
> 2. Our comment about Ox (outcome explanation engines) being slower than Mx (model explanation engines) is general, because, most of the existing techniques in Ox, like lime [3], learn a different auxiliary explainable model for “each” test prediction separately whereas, in Mx, only one global explainable auxiliary model is learnt. To the best of our knowledge, we are the first ones to create an outcome explanation engine for the task of KBC and we take a different approach than learning a new explainable model for each test prediction separately.
> To explain a prediction in OxKBC, most of the time is spent in extracting features for different templates. For FB15k, it takes on average of around 500 milliseconds per test sample to extract template features using a single core of an Intel(R) Xeon(R) W-2145 CPU @ 3.70GHz. Once features are extracted, it takes only 0.38 milliseconds to select the best template using the selection module. On the other hand, in rule mining, once the rules are mined, it takes only 2.04 milliseconds to select the best rule for a given test sample. Even though at test time, OxKBC is much slower than rule mining, we believe it is still fast enough to be usable, especially for a human interacting with an explanation interface. We also note that this extra inference time comes with the advantage of improved quality of explanations.
>
> 3. Yes, we observe a very high standard deviation for the unsupervised setting. This is primarily because it is very difficult to train the model using only distant supervision in the form of unsupervised loss term. As a result, in a couple of random runs, it learns a degenerate model where all predictions collapse randomly to any one of the templates. Adding KL divergence term along with a small amount of labeled data (80 samples) in the overall loss term helps in maintaining the same posterior distribution over the output space and ensures that it doesn’t learn a degenerate model, thus reducing the variance.
>
> 4.  The supervised training is done using only 80 data points and hence may not achieve as high performance as in a semi-supervised setting which uses the entire training data as unlabeled data along with manually annotated 80 labeled data points. Our semi-supervised learning also includes additional evidence using the KL divergence term that is computed over unlabeled data.  Overall, this is inline with general observation of semi-supervised learning outperforming purely supervised learning, by the way of exploiting statistical regularities present in large amounts of unlabeled data.
>
> 5. Since the amount of labeled data is very less (80 data points), we may not see any trend by changing the amount of supervision. Nevertheless, we trained our model for FB15k using 1/3rd (27 data points) and 2/3rd (56 data points) of the labeled data and achieved 54.8% and 60% F1 scores respectively.
>
> We will do our best to incorporate all the clarifications in the main paper if the space permits or alternatively in the supplementary text. We have sent an email to the program chairs asking if we are allowed an extra page to address the reviewers’ comments and are waiting for their response.
>
> [1] R. Das et al. Go for a walk and arrive at the answer: Reasoning over paths in knowledge bases using reinforcement learning. In ICLR, 2018
> [2]  P. Jain et al.  Type-sensitive knowledge base inference without explicit type supervision. In ACL, 2018.
> [3] M. T. Ribeiro et al. "why should I trust you?": Explaining the predictions of any classifier. In KDD, 2016

---

### Official Review · AnonReviewer2 · 2020-03-27
**Generating explanations of factorization models for link prediction with template scoring**

**Rating:** 7
**Confidence:** 3

**Review:**

The authors propose a novel method OxKBC for generating post-hoc explanations of a trained factorization model for link prediction. The authors identify five different templates which explain a predicted triple (s,r,o): Similarity between tail entities, similarity between relations, both occurring simultaneously, similarity between r and the hadamard product of a two-hop relation in the KB, and o being a frequent tail entity for r. Since the predefined scoring functions do not obviously generalize between different templates, OxKBC performs a two-step computation for each predicted triple where first the best template is selected, and second that template is grounded in the knowledge graph. The authors evaluate OxKBC using a paired comparisons test with Amazon Mechanical Turk, comparing against a rule-mining baseline. Results show OxKBC to handily outperform the baseline.

The method presented in the paper is interesting and, to the best of my knowledge, novel. The results are strong, the subject matter is interesting and timely, and I would like to see a version of this paper at AKBC. With that said, there is a somewhat significant design flaw in the model which should, at least, be discussed in the final version: The paper aims to explain the predictions of an existing factorization through the selection of a "best" instantiation of a template. However, due to the nature of factorization models, there is no guarantee that any prediction is explaining by a single instantiation, and not a disjunction or conjuction of instantiations, and OxKBC has no mechanism for identifying or addressing such cases. In other words, there may be two instantiations which are *equally good*, or two instantiations which only functions as an explanation *when both are present*, and in these cases all OxKBC can do is score each instantiation.

Other comments & questions:

- Is there a reason why "entity & two-length relation similarity" does not appear as a sixth template? It seems like an obvious continuation of the pattern of the existing templates.
- The notation of T_i for a score of a template and Ti for the template being scored in Equation 2 is a bit confusing. I would suggest using another symbol to differentiate the two.

---

> ### Author Response · Authors · 2020-04-06
> **Reply to AnonReviewer2**
>
> Dear Reviewer,
>
> Thanks for your valuable comments and suggestions. Below we have tried our best to address each of your concerns:
>
> Q1:  "There is a somewhat significant design flaw in the model which should, at least, be discussed in the final version: The paper aims to explain the predictions of an existing factorization through the selection of a "best" instantiation of a template. However, due to the nature of factorization models, there is no guarantee that any prediction is explained by a single instantiation, and not a disjunction or conjunction of instantiations, and OxKBC has no mechanism for identifying or addressing such cases. In other words, there may be two instantiations which are *equally good*, or two instantiations which only functions as an explanation *when both are present*, and in these cases all OxKBC can do is score each instantiation."
>
> A1: You make an important point. Let us first distinguish between two cases: (I) alternative independent explanations (disjunctions), and (II) multiple paths that collectively construct a single explanation (conjunctions). We believe that OxKBC can be extended to both these cases with some effort, though the approaches will be different.
>
> Case I (Disjunctions): OxKBC currently outputs the top explanation, i.e., the top grounding of the selected template. To generate alternative explanations, it can always generate the next best groundings of the selected template. However, alternative good explanations may also come from a different template. This will require us to learn an explanation model that can compare the second best grounding of a top template with the best grounding of the second best template (and so on). More thought will be needed on how to compare these. It would have been easier if we were given explanation training data, but notice that our design of OxKBC assumes minimal supervision.
>
> Case II (Conjunctions): Sometimes multiple paths may collectively construct a longer multi-step path. This can be handled in our current formulation by defining new (longer) path templates. However, to handle its general form, we need to generalize the notion of explanation as a “path” to explanation as a “subgraph”. We believe that we can easily extend our model for such explanations. We just need to define new rules within the explanation grammar to allow for general subgraphs. We can then define a new template, and provide a Compatibility function for the template. The selection module should follow as is. For instance, (X, is Spouse of, Y) because (X, is father of Z) and (Y, is mother of, Z) could be an instance of some template where  P <- P1 and P2, head(P1) = head(P), head(P2) = tail(P), tail(P1)=tail(P2). We did not study this in depth because, in our preliminary analysis on our datasets, paths came out as natural explanations, and more generic subgraphs were not found essential.
>
>
> Q2: "Is there a reason why "entity & two-length relation similarity" does not appear as a sixth template? It seems like an obvious continuation of the pattern of the existing templates."
>
> A2: One can definitely define a sixth template for “entity & two-length relation similarity” as a new template.
> We presented only those templates in the paper which were used in our preliminary investigation of potential explanations for our datasets.
>
> Q3: The notation of T_i for a score of a template and Ti for the template being scored in Equation 2 is a bit confusing. I would suggest using another symbol to differentiate the two.
>
> A3: Thanks for the feedback. We will use a different symbol for T_i in the final version.
>
> We will do our best to incorporate all the clarifications in the main paper if the space permits or alternatively in the supplementary text. We have sent an email to the program chairs asking if we are allowed an extra page to address the reviewers’ comments and are waiting for their response.

---

### Official Review · AnonReviewer3 · 2020-03-28
**Effective approach to explaining tensor factorization models's prediction for KB completion task**

**Rating:** 7
**Confidence:** 4

**Review:**

The paper proposes a novel approach to providing an explanation for the prediction made by off-the-shelf tensor factorization models by using its scores to create an augmented weighted knowledge graph and scoring higher order paths (aka templates) as explanations. The presentation is mostly clear and the results are convincing of the proposed idea. The paper however suffers from lack of sufficient novelty and adequate experimental comparison with other state-of-the-art approaches that offer explanations.

Cons:
[Novelty] The paper considers path-based approach to providing explanations to predictions of a KBC model. This idea of using a path is not new. In fact, the similarity function in equation 1 is very similar to KL-REL (Shiralkar et al.). PRA (Lao et al.) and PredPath (Xi et al.) is another paper that has considered meta-paths (same as templates in current paper) for ranking explanations.

[Weak baseline and lack of adequate experimental results] The baseline of rule mining seems to be old and a weak one. Although the proposed approach is meant to be faithful to its TF model, since it is a path based approach to providing explanations, it might be worth comparing it to approaches that work with the observable graph. The rule mining approach considered in the paper is appropriate, but an old one and approaches such as KL-REL (single path by Shiralkar et al.) have been proposed that are promising. It might be worth comparing to such recent explainable models. Secondly, some discussion around why/how an explanation provided by the path derived by proposed approach might correlate with statistical information summarized by the TF model will be useful.

[Formulation] The underlying TF model draws upon the global, long-range statistical knowledge to derive the prediction for an example. how can this prediction be explained by a single explanation path? In practice, facts can often be explained by alternative paths and/or multiple paths that collectively provide evidence and may fail to justify the prediction individually. E.g. (X, isSpouseOf, Y) can be explained by the fact that they have a child together, or X is son-in-law of Z and Z is mother of Y. Some discussion regarding this bias to use single path is missing.

---

> ### Author Response · Authors · 2020-04-06
> **Reply to AnonReviewer3**
>
> Dear reviewer,
> Thanks for your valuable comments and suggestions. Below we have tried our best to address each of your concerns:
>
> [Novelty]: Thank you for pointing us to similar work like KL-REL and PredPath. We will cite them appropriately in the paper.
> We agree that path based approaches are not new. The novelty of our approach stems from the way we formulate our task, the high level solution, and some specific technical details. In particular: (1) to the best of our knowledge, we present the first outcome explanation system faithful to an underlying tensor factorization model for a KBC task, (2) our specific way to divide the problem is also novel -- training a second-order template selection module, and then grounding it to generate the explanation, (3) augmentation of KG with entity similarity edges, which can be viewed as representing an aggregation of multiple 2-length paths, (4) use of a neural model for template selection, with a novel unsupervised loss function; we also construct good features for each template. In addition, our experimental demonstration shows that stable training can be achieved with minimal supervision, and that human users find explanation plausible compared to existing baselines.
>
> [Weak baseline]: Thanks for pointing us to the KL-REL approach, we will cite it appropriately in our paper.
> As you rightly pointed out, our goal was to generate an explanation for the prediction of the underlying TF model and at the same time remain faithful to it. We compared it with a variant of rule-mining proposed in [1], which is the only approach we were aware of that mines rules based on the underlying TF model. We checked the KL-REL paper -- it's a fact-checking technique that augments KL with relation similarity defined in the paper using the contracted dual graph. It can be used to generate an explanation path for a given prediction of the TF model, but it is not clear why the explanation would be faithful to it. It may be interesting to combine KL-REL ideas with statistical information in TF models. We will explore this direction in the future.
>
> [“Secondly, some discussion around why/how an explanation provided by the path derived by proposed approach might correlate with statistical information summarized by the TF model will be useful.”]:
>  We shall add a note in the paper (if an extra page is available), or in a supplementary section to discuss the same:
>
> Multiplicative models like TypeDistMult use three-way products to compute scores of a triple. So, a high dot product (and therefore, cosine score) between two relation embeddings r and r’ (or two entity embeddings) fundamentally represents that model M considers the two relations (entities) as somewhat replaceable. This idea is exploited in our relation similarity template T1. Consider, as an example, a KBC query r(s,?) and the corresponding prediction, o, for the tail entity by the model M. Say OxKBC explains the prediction using T1 by finding the best fact r’(s,o) in the KB such that r ~ r’ (cosine score between the two are high). Now, since r’(s,o) is in the KB, model score for (s,r’,o) must be trained to be high, and hence, Hadamard product of the embeddings of s and o is likely to be closely aligned with the embedding of r'. Because r ~ r’ Hadamard product of the embeddings of s and o is also closely aligned with that of r, resulting in a high model score for (s,r,o). Thus, this explanation is one reasonable clue why model M may have decided to make this prediction in the first place. Similar arguments can be made for T2, T3 and T4.
>
> [Formulation]:  You make an important point. Let us first distinguish between two cases: (I) alternative independent explanations, and (II) multiple paths that collectively construct a single explanation. We believe that OxKBC can be extended to both these cases with some effort, though the approaches will be different.
>
> Case I: OxKBC currently outputs the top explanation, i.e., the top grounding of the selected template. To generate alternative explanations, it can always generate the next best groundings of the selected template. However, alternative good explanations may also come from a different template. This will require us to learn an explanation model that can compare the second best grounding of a top template with the best grounding of the second best template (and so on). More thought will be needed on how to compare these. It would have been easier if we were given training data on multiple explanations per prediction, but notice that our design of OxKBC assumes minimal supervision.
>
> Case II: Contd. in the next box...
> ....
> [1] Bishan Yang, Wen-tau Yih, Xiaodong He, Jianfeng Gao, and Li Deng. Embedding entities and relations for learning and inference in knowledge bases. In ICLR, 2015

---

> > ### Author Response · Authors · 2020-04-06
> > **Reply to AnonReviewer3 contd.**
> >
> > In continuation of the previous response:
> >
> > Case II: Sometimes multiple paths may collectively construct a longer single multi-step path. This can be handled in our current formulation by defining new (longer) path templates. However, to handle this in its general form, we need to generalize the notion of explanation as a “path” to explanation as a “subgraph”. We believe that we can easily extend our model for such explanations. We just need to define new rules within the explanation grammar to allow for general subgraphs. We can then define a new template, and provide a Compatibility function for the template. The selection module should follow as is. For instance, (X, is Spouse of, Y) because (X, is father of, Z) and (Y, is mother of, Z) could be an instance of some template where  P <- P1 and P2, head(P1) = head(P), head(P2) = tail(P), tail(P1)=tail(P2). We did not study this in depth because, in our preliminary analysis on our datasets, paths came out as natural explanations, and more generic subgraphs were not found essential.
> >
> > We will do our best to incorporate all the clarifications in the main paper if the space permits or alternatively in the supplementary text. We have sent an email to the program chairs asking if we are allowed an extra page to address the reviewers’ comments and are waiting for their response.

---

### Author Response · Authors · 2020-04-17
**Revision incorporating reviewers' comments**

Dear reviewers,

Thank you once again for your valuable feedback. We have incorporated all your suggestions/comments and have uploaded a revised paper clarifying your concerns to the best of our ability. Please let us know if any of the concerns still remain unanswered and we will be more than happy to address that.

Regards,

---

### Decision · Program_Chairs · 2020-04-30

**Decision:**

Accept

**Comment:**

The paper looks into explaining the predictions of factorization models (i.e. post-hoc interpretation). This is an important problem, and there is very little work into this in the context of factorization models.

Judging from the paper and the authors' replies here in the discussion forum, the goal is to produce explanations faithful to the underlying factorization model. However, there is between the goal and the evaluation, and certain confusion in the discussion. If the goal is to understand what a factorization model relies on,  it is unclear that it makes sense to train the model on human-annotated data or measure how well it agrees with explanations given by humans. We would like to understand what the model is doing rather than what a human thinks the model should be doing.  It may well rely on some obscure artifacts, and we would like to know this.

If the goal is generating plausible explanations, then it opens up the paper to criticism by one of the reviewers -- there should be a broader comparison to 'more explainable' models, which are not tied to factorization models.

Overall,  even given these issues with discussion and maybe certain overselling of its faithfulness (even though the explanation on page 6 is plausible, it is still a heuristic), I like the work.  It is hard to evaluate faithfulness and there are interesting insights in this submission. Also, this criticism applies to quite a few published papers in interpretability.

We expect the author's put some effort into sharpening the argument and addressing AC and reviewers' concerns.  I would have loved to see a discussion of stability, maybe artificial experiments where we know what the model is doing, etc.